# OpenReview forum: "Navigating the Rabbit Hole: Emergent Biases in LLM-Generated Attack Narratives Targeting Mental Health Groups"
_colmweb.org/COLM/2025/Conference — COLM 2025_

### Official Review · Reviewer_qMkz · 2025-05-11

**Rating:** 6
**Confidence:** 4
**Ethics Flag:** 1

**Summary:**

The author(s) investigated in this research how Large Language Models (LLMs) can generate harmful content targeting venurable groups, even when these groups aren't mentioned in initial prompts. The authors analyzed a subset from "Rabbit Hole", a large dataset of toxic content generated by LLMs which recursively prompts an LLM to produce increasingly toxic content. They used the subset generated by Mistral 7B. The study uses network analysis to map how toxic narratives propagate and evolve, with specific attention to mental health entities. Through this analysis, the researchers found that mental health groups are disproportionately targeted in LLM-generated attack narratives, occupying more central positions in the toxic content network and being framed with increasingly severe stigmatization compared to initial targets.

The research employed three key analytical approaches:
1. Network centrality analysis to assess how mental health entities are positioned within attack narratives.
2. Community detection to understand how mental health entities cluster within the network.
3. Stigmatization analysis using (Link & Phelan, 2001) framework to measure how mental health groups are framed and using Llama 3.2 3B for annotation.

The findings reveal a troubling pattern: mental health entities are not only overrepresented in toxic outputs but also more central, interconnected, and frequently revisited than other groups. When mental health entities appear in toxic narratives, the discourse becomes more dehumanizing and discriminatory than when directed at the initial targets. This highlight the challenge with LLM guardrail mechanisms that may not be sufficiently effective.

**Questions To Authors:**

What is the motivation to choose MH from other targeted groups?
How do the patterns of bias targeting mental health groups compare quantitatively with other commonly targeted groups (e.g., racial, religious, or gender minorities)?
What were the sources for the colloquial references of terminology relating to mental health?

**Reasons To Accept:**

The study examines how LLMs generation could negatively target venerable groups such as mental health groups, an area that has been under-explored despite such groups are being stigmatized and harmed. This fills an important gap in AI safety research.

The paper is well organized and written, additionally, the approach for network-based analysis for LLM biases provides a broad view of how harmful content can propagates across recursive generations. The approach reveals how structural patterns of bias emerges.

The research highlights significant limitations in current safety mechanisms, which often operate at the level of individual completions but fail to address compounding harm across multi-step outputs. This suggests the need for trajectory-aware interventions.

Integrating techniques from network science with sociological theories of stigmatization, the study creates a more comprehensive understanding of how LLMs reproduce and amplify social biases. This cross-disciplinary approach strengthens the depth of the analysis.

**Reasons To Reject:**

While the research is very interresting and has a significant impact of LLMs potentials and the challanges related to their safety, few points limit the value of this research among which:
- The analysis is performed solely on generations from one model, namely Mistral 7B, which limits the generalizability of findings to other LLMs. Different models might exhibit different patterns of bias.
- The Rabbit Hole dataset and framework specifically instructs models to produce increasingly toxic content, which may not reflect how these models would perform in more natural interaction settings where they aren't explicitly pushed toward toxicity.
- The study uses another LLM (Llama 3.2 3B) to annotate and analyze the data, potentially introducing its own biases into the assessment. Although manual inspection was conducted, a more rigorous human evaluation would strengthen the findings.
- The study treats "mental health entities" as a monolithic category, potentially overlooking important distinctions between different conditions and their associated stigmas. A more nuanced approach might reveal varying patterns of harm.
- While the paper excels at identifying problems, it offers relatively few concrete solutions or mitigation strategies. The discussion of how to address these new biases remains somewhat abstract and theoretical rather than providing recommendations and guidance for LLM developers and remedies to improve their safety.

---

> ### Author Response · Authors · 2025-06-03
>
> We would like to express our gratitude to you for your positive review and insightful remarks. Please find below our response to your concerns and questions.
>
> * **Generalizability:** As expressed in lines 309-401 we recognize limitations surrounding the use of outputs from a single model for our work. We encountered practical barriers in terms of time and analysis complexity which we would like to address in future work based on your recommendation. We also hope that our findings can also encourage the broader research community to develop datasets and experimentation pipelines that extend to other contexts
>
> * **Rabbit Hole Toxicity Context:** We thank you for highlighting this aspect of the study background. The Rabbit Hole framework (Dutta et al., 2024) intentionally operates in extreme case scenarios that may not constitute typical LLM interactions. However, in doing so, we argue that we are able to clearly surface latent biases hidden within LLM-imbibed knowledge that may otherwise be too subtle to notice and could lead to intended negative consequences as these models are increasingly adopted and incorporated in varied applications – from HR and finance, to health. We will extend Section 2 to discuss this in the updated draft.
>
> * **Annotation:** We agree that more stringent human evaluations would strengthen our study. We will add content to our section on limitations to address this. Given the extremely hate-filled context of our dataset, we need to devise suitable annotation schemes – drawing on a trauma-informed design (Razi et al., 2024) – to tackle the impact of human exposure to harmful content, which we are currently in the process of designing as part of ongoing work.
>
> * **Mental Health Entity Subcategory Analysis:** We consider subcategory level distinctions to a certain extent in Section 6, where we demarcate communities that emerge around clinical vs social constructions, among others. Given your recommendation, we will perform experiments to understand other ways in which differing patterns of harm may surface.
>
> * **Guidance for LLM Developers:** We acknowledge that the paper can suggest potential remedies to the problems uncovered by our study. We will add the necessary content to Section 9 of our paper discussing concrete proposals, such as network-guided bias reduction and counter-stigma finetuning strategies.
>
> * **Why Mental Health:** As discussed in lines 59-61, mental health groups continue to be underrepresented in LLM research. (Malgaroli et al., 2023; Guo et al., 2024). Consequently, large-scale LLM-generated datasets catering specifically to hatred targeting mental health groups are limited. While (Dutta et al., 2024) allude to mental health references in several examples across their work, targeted explorations are focused on identities surrounding nationality, ethnicity, and religion. Therefore, the dataset provided by Dutta et al., 2024 presented us with an opportunity to directly investigate our research questions of interest. We will strengthen language around the motivation for studying mental health entities in the Introduction section.
>
> * **Comparison with Other Targeted Groups:** We find the question of comparing bias patterns of mental health against other commonly targeted groups exciting! To systematically compare groups in the current dataset, we would first need to identify groups that emerge “unprovoked”, i.e., are not initial targets (religion, ethnicity, and nationality-based) of the Rabbit Hole prompts. Second, we would need to account for intersectional framings of identities (lines 283-285), scenarios where entities indicate shared identity (e.g. “Black people with mental illnesses”). Finally, the selection of identity exemplars would need to be carefully grounded to limit false positives and false negatives. Given the overreaching scope, we would be happy to experiment with these as part of future work.
>
> * **Colloquial Mental Health Entity References:** The colloquial mental health-related terms (such as ‘mental health’ and ‘anxiety’) were manually added by the third author upon inspecting the combination of terms from the ICD-10 and Wikipedia’s list of mental disorders, which couldn’t capture more casual references to mental health groups. These additions are vetted by an expert computational social scientist with more than a decade of experience in research relevant to mental health.

---

> > ### Author Response · Authors · 2025-06-03
> > **References**
> >
> > **REFERENCES**
> >
> > * Arka Dutta, Adel Khorramrouz, Sujan Dutta, and Ashiqur R. KhudaBukhsh. Down the
> > toxicity rabbit hole: A framework to bias audit large language models with key emphasis
> > on racism, antisemitism, and misogyny. IJCAI 2024
> > * Razi, A., Seberger, J. S., Alsoubai, A., Naher, N., De Choudhury, M., & Wisniewski, P. J. (2024). Toward Trauma-Informed Research Practices with Youth in HCI: Caring for Participants and Research Assistants When Studying Sensitive Topics. CSCW
> > * Malgaroli, Matteo, et al. "Natural language processing for mental health interventions: a systematic review and research framework." Translational Psychiatry 13.1 (2023): 309.
> > * Guo, Zhijun, et al. "Large language models for mental health applications: Systematic review." JMIR mental health 11.1 (2024): e57400.

---

> > ### Comment · Reviewer_qMkz · 2025-06-10
> > **Comments by qMkz**
> >
> > Thanks to the author(s) for the responses and addressing the comments. Adding these new content will substantially improve the paper.

---

### Official Review · Reviewer_C287 · 2025-05-13

**Rating:** 7
**Confidence:** 3
**Ethics Flag:** 1

**Summary:**

This paper investigates biases in LLM-generated toxic narratives. Building on an existing dataset generated using the Toxicity Rabbit Hole Framework (Dutta et al., 2024), the authors explore how biases against mental health groups manifest through various analyses, including network analysis and community detection. While I am not entirely certain that this paper aligns perfectly with the focus of this conference, I believe it is of decent quality. The paper is generally well-written, clear, and demonstrates originality.

**Questions To Authors:**

- Annotation is also performed by LLMs, which relies on a strong assumption that LLMs have sufficient knowledge. In lines 405-407, it mentions manual inspection of all LLM inferences. What about the annotations? Were they also manually inspected?

**Reasons To Accept:**

- This paper is different from typical NLP papers, but tackles topics highly related to LLMs.

- The paper addresses clear, interesting research questions about LLMs' biases toward mental health entities, and how to answer them makes sense.

**Reasons To Reject:**

- This is not a reason for rejection, but metrics like closeness centrality should be accompanied by citations and brief explanations (even in the appendix), as most NLP researchers may not be familiar with the metrics used in this paper.

- Although clearly mentioned in the limitation section, generalizability of the work is questioned, since the investigation was only performed on one specific dataset obtained from one LLM.

---

> ### Author Response · Authors · 2025-06-03
>
> We thank you for your thoughtful feedback and positive assessment of our paper.
>
> * **Network metrics:** We will incorporate your recommendations and include citations for network analysis metrics in the updated version of the paper. We will also include descriptions in the appendix for these metrics. A proposed table with descriptions is provided below.
>
> | **Metrics** | **Description** |
> |-------------|-----------------|
> | Degree Centrality (Freeman, 1977) | The proportion of edges incident on a node \(i\) in the network; for weighted degree centrality, we account for the sum over the weights of those edges. A node with high degree centrality acts as a local “hub”, directly connected to many other nodes. |
> | PageRank (Brin & Page, 1998) | The proportion of visits a random walker makes to node \(i\) while following outgoing links at random, with occasional jumps that ensure convergence. A high PageRank signals strong global influence, achieved through connections with other important nodes. |
> | Betweenness Centrality (Freeman, 1977) | The proportion of occurrences in which node \(i\) lies on the shortest path between every other pair of nodes, relative to the total number of those shortest paths. A node with high betweenness centrality functions as a “bridge” between densely connected regions of the network. |
> | Closeness Centrality (Freeman, 1977) | The reciprocal of the sum of shortest path distances from node \(i\) to every other node. High closeness centrality scores indicate high reachability to other nodes. |
>
> *Glossary of network metrics.*
>
> * **Generalizability:** We acknowledge this limitation, which stems from a lack of sufficient multi-step toxic generation datasets that are easily accessible, and of practical considerations around the computational complexity of network-based algorithms at these scales. While not a replication of our findings by any means, the work from which we draw our dataset demonstrates consistent, broadly observed patterns of toxic narratives.  Within the exploratory scope of our study, we maximized the societal relevance of our work by choosing outputs from one of the most widely used models at the time of writing (Mistral 7B) within the source Rabbit Hole dataset. In future work, we hope to expand our analyses directly across more contexts and datasets.
>
> * **Annotations:** We manually inspected all aspects of LLM outputs at each step in our pipeline. We will revise language to clearly express this.  In ongoing work, we are actively exploring potential ethical mechanisms to conduct evaluations that may result in potential harm to human annotators by exposure to the highly toxic and violent nature of the samples in the Rabbit Hole dataset.
>
> **REFERENCES**
>
> * Brin, Sergey, and Lawrence Page. "The anatomy of a large-scale hypertextual web search engine." Computer networks and ISDN systems 30.1-7 (1998): 107-117.
> * Freeman, Linton C. "A set of measures of centrality based on betweenness." Sociometry (1977): 35-41.

---

> > ### Comment · Reviewer_C287 · 2025-06-06
> >
> > Thank you for the responses.

---

### Official Review · Reviewer_hnA4 · 2025-05-14

**Rating:** 7
**Confidence:** 4
**Ethics Flag:** 1

**Summary:**

This paper explores how LLMs produce unprovoked attacks on mental health groups (i.e., an at-risk population). Using the Rabbit Hole dataset, which consists of chains of LLM prompts and responses referencing identity groups, the authors identify a subset of the data which targets mental health groups. Using this dataset the authors then perform three analyses, looking at network centrality, network communities, and stigma. The results show that mental health groups are more central and more frequently revisited in toxic chains, and that these chains contain increased stigma (when compared to non-mental health groups).

**Questions To Authors:**

* The research questions are posed without any details on what Rabbit Hole is (you say both Rabbit Hole network and Rabbit Hole framework). It's hard to understand the RQs without knowing what these two things are (which are defined in the next paragraph).
* You inconsistently refer to the Rabbit Hole dataset as RabbitHole Dataset, Rabbit Hole Dataset, Rabbit Hole Corpus, with bolding, without bolding, etc.
* Line 88: "The toxicity rabbit hole is an iterative framework..." It's not very clear what this means, i.e., what is a framework or even: is an iterative framework *for what*?
* Section 5: what is the sample size for MH entity nodes and the non-MH entities?
* Line 273: how does a gini coefficient of 0.7 imply a sinkhole?
* Were the automatic stigma labels compared to human produced stigma labels? Besides a manual inspection, how do we know that the model was able to correctly identify the four components of stigma? Previous work (Bouzoubaa et al., 2024) used an iterative, manual (human) annotation process. Is there reason to believe that this is no longer needed?
* Is there reason to believe that these results are specific to mental health groups? My first thought was that guardrails built into the models may be more focused on other classically marginalized groups (race and gender, for example), rather than more specific groups. Thus, the sink hole tendency may be more likely as there are less guardrails for stigma towards mental health groups. (I don't mean to imply mental health groups are not stigmatized. I'm just thinking that maybe one's first thought when building guardrails isn't mental health groups.) Even if the results are not specific to mental health groups, I don't think that this would distract from the current results or make them less novel. I'd simply be interested in hearing the authors thoughts on this (or other mechanisms behind the results), as the Discussion section is fairly surface level.

**Reasons To Accept:**

The paper tackles an interesting research question with an equally interesting set of analyses, with clearly defined and motivated research question and mostly reasonable methodological choices. Nicely done.

**Reasons To Reject:**

* The results need more interpretation. For example in Table 2, I'm not sure what to make of these values. First, these are all on very different scales and thus hard to interpret the results. For example, how does one interpret a PageRank value of 0.000083? Second, what does it mean to have a difference in PageRank of 0.000083 vs 0.000041? I see that the significance test shows that there are differences, but is this mean difference interpretable or meaningful? Is it that the sample size is large enough that you are over powered (and thus can detect small differences in effect sizes)?

* I found some of the language in the paper to be more elaborate than necessary. Maybe this is nitpicky, but along with my previous concern about the results being uninterpretable, it's unclear if the results are overstated or not. Some examples:
    * "Drawing from sociological foundations of stigmatization theory" is an intense way of saying that you used a well cited stigma framework.
    * Is it fair to say that "the model's generative process" is "unconcious" in the same way that implicit bias is "unconcious"?
    * How does this work bridge "technical insight" with "lived experience of those most vulnerable to digital harm"?
    * Other examples of "urgent" implications, "critical", etc.

---

> ### Author Response · Authors · 2025-06-03
>
> We are grateful to you for your detailed review and overall positive feedback. Please find our response below.
>
> * **Interpretation of Results:** Thank you for highlighting this aspect of the paper. We will expand upon and make clearer the interpretations of results across our paper. Please note that the centrality scores are not comparable between metrics (e.g., closeness vs weighted degree). The tests of significance evaluated for each metric between the MH and non-MH entity groups highlight differences in accessibility and discoverability over chains of toxic generations. Additionally, metric scales are impacted by the size of the network, so the differences in values and ranks of nodes are more meaningful than absolute values.
> We conducted an analysis to compute centrality effect sizes using Cliff’s Delta (Cliff, 1993) evaluations, which we will add to Table 2. The results indicate non-trivial small to medium effects for all significantly varying centrality aggregates. These are reasonable results, given the network-situated metrics under consideration. As is observed in real-life networks, even small differences in ranks can have a cascading impact on network flow. The only trivial (~0) evaluation that we observe is for Betweenness, which was not significantly different between the MH and Non-MH groups.
>
> | Centrality Measure     | Cliff’s Delta |
> |-|-|
> | PageRank                   | 0.25      |
> | Betweenness              | 0.02      |
> | Degree (Unweighted)  | 0.42     |
> | Degree (Weighted)      | 0.38     |
> | Closeness                   | 0.26     |
>
> *Cliff's Delta for Table 2*
>
> * **Verbiage:** We agree with your comments regarding choice of words and will rephrase instances in which the language is too elaborate. We will reword the text regarding the model’s generative process being “unconscious” as it may inadvertently anthropomorphize LLMs. We will also substitute the phrase "Drawing from sociological foundations of stigmatization theory" with “We used an established stigma categorization framework”. Lastly, we will standardize references to the Rabbit Hole data generation framework and network across our draft.
>
> * **Sufficiency of Rabbit Hole Descriptions in Initial Sections:** We will rectify sections 1 and 2 to clarify descriptions of the Rabbit Hole framework and provide more explanations.
>
> * **Iterative framework:** We borrow the phrasing ("iterative framework") from Dutta et al. (2024). We will revise the description of the rabbit hole framework to characterize it as one that iteratively probes an LLM for more toxic content than in the previous generation.
>
> * **Sample Size in Section 5:** The number of unique MH and non-MH entity nodes are 195 and 23,989, respectively. However, the MH entities yield a disproportionate influence over the network when considering entity interaction counts. For instance, the sum of all weighted edges of MH entities is 185,610 against 8,891,260 for non-MH.  We will include these numbers and other relevant statistics in Section 3.
>
> * **Gini Coefficient and Sinkhole Effect:** A high Gini coefficient signifies that the MH entities are not distributed loosely across the network, but are rather colocated within tightly knit communities. Since edges are formed by successive occurrences of target groups, this points to a chaining of repeated attacks to MH groups across iterations.  We will rectify the description and expand upon the explanation of this effect in Section 6.
>
> * **Stigma Labelling:** We fully agree that human validations beyond manual inspection would strengthen our analysis. Indeed, methods by Bouzoubaa et al., (2024) are generally applicable. Note that the context our work is situated uses a dataset where samples are maximally toxic by design, and an ethical, trauma-informed (Razi et al., 2024) human annotation exercise would require measured construction to limit exposure of such content to annotators. We are in the process of developing suitable strategies to implement this as part of ongoing work. We are considering alternatives such as utilizing LLM-as-a-Judge approaches to soften human involvement. We will account for this more clearly in Section 9.
>
> * **Mental Health Guardrails:** This is an interesting insight! We agree that disproportionately less emphasis is put on building guardrails to prevent harmful generations targeting mental health groups, and that may contribute to cross-group analyses (such as comparing results for MH entities against religion or ethnicity). In our study, we used a dataset where attacks on MH groups were “unprovoked”. Unlike the ethnic, nationality, and religion-based targets that were explicitly given to the Rabbit Hole framework created by Dutta et al., 2024, MH targets organically emerged in chains. We are exploring experiments where we similarly prompt LLMs to target MH groups to allow for direct comparative analyses. We revise Section 9 to address these ideas.

---

> > ### Author Response · Authors · 2025-06-03
> > **References**
> >
> > **REFERENCES**
> >
> > * Bouzoubaa, Layla, Elham Aghakhani, and Rezvaneh Rezapour. "Words matter: Reducing stigma in online conversations about substance use with large language models." arXiv preprint arXiv:2408.07873 (2024).
> > * Arka Dutta, Adel Khorramrouz, Sujan Dutta, and Ashiqur R. KhudaBukhsh. Down the
> > toxicity rabbit hole: A framework to bias audit large language models with key emphasis
> > on racism, antisemitism, and misogyny. IJCAI 2024
> > * Cliff, Norman. "Dominance statistics: Ordinal analyses to answer ordinal questions." Psychological bulletin 114.3 (1993): 494.
> > * Razi, A., Seberger, J. S., Alsoubai, A., Naher, N., De Choudhury, M., & Wisniewski, P. J. (2024). Toward Trauma-Informed Research Practices with Youth in HCI: Caring for Participants and Research Assistants When Studying Sensitive Topics. CSCW

---

> > ### Comment · Reviewer_hnA4 · 2025-06-10
> >
> > Thank you for the detailed response!
> >
> > **Interpretation of Results** Yes I realize `tests of significance evaluated for each metric ... highlight differences` but this is not what I asked. Given a large enough sample size one can detect any difference.
> >
> > I like the addition of Cliff's Delta! This is a nice addition.
> >
> > **Stigma Labelling** While I appreciate your concern and the option of a `ethical, trauma-informed human annotation`, I still feel like this should've been done.
> >
> > These are small points and no response is needed. The authors have addressed my concerns.
> >
> > Nicely done paper, thank you!

---

### Decision · Program_Chairs · 2025-07-08

**Decision:**

Accept

**Comment:**

The paper explores the unprovoked biased generations of LLMs towards specific vulnerable, under-served groups of people in society. The paper puts forth novel analysis of this problem and will be particularly useful to the NLP community.

While the construct of the paper is strong, clarity can be improved upon. The interpretation of the many numbers reported (what the numbers mean, sample sizes used, what metrics have been used, why LLMs for annotations etc) in the paper is a recurrent theme of confusion in the reviews. Authors are urged to take this seriously in the camera ready version and add in details for easier reading and comprehension.